# pGS-CAM: Interpretable LiDAR Point Cloud Semantic Segmentation via Gradient Based Localization

**Abhishek Kuriyal, Vaibhav Kumar**
Department of Data Science and Engineering
Indian Institute of Science Education and Research, Bhopal
`{abhishek19, vaibhav}@iiserb.ac.in`

## Abstract

To extract the local information required for effective semantic segmentation of point clouds, a number of deep learning architectures typically make use of sophisticated feature extractors. Unfortunately, there has not been a lot of discussion on how to interpret their forecasts, which is essential if deployed in real-world settings. To that end, we propose pGS-CAM (point cloud Grad-Seg-CAM), a quick and effective gradient-based method for class activation mapping in point cloud semantic segmentation architectures. To gain insight into what each intermediate layer of the architecture does, our technique provides a heatmap for the corresponding layer. We use the popular semantic segmentation architecture (RandLA-Net) and a commonly used MLS dataset (SemanticKITTI) for our experimentation.

## 1 Introduction

Various applications, including autonomous vehicles, require perception and interpretation of the surroundings captured using Light Detection and Ranging (LiDAR)-based sensors. These sensors generate outcomes in the form of point clouds. A lot of research has been done on developing effective semantic segmentation architectures for point clouds Xie et al. (2020); Bello et al. (2020). However, a crucial component of these models, i.e., interpretability, still remains unexplored. An intricate understanding of these models if they are deployed in real-world systems is required for better development. One way of achieving this is by visually indicating the regions of point clouds that are influencing the learning of a neural network model and its decisions by creating *heatmaps* for each activation layer of the network. This further aids in network design and reflects the changes caused by the addition of each layer.

Recently, some attempts have been made to evaluate the point-wise importance of instance classification and object detection tasks. Zheng et al. (2019) conducted point dropping by shifting points towards the point cloud's centroid to assess each point's importance in the classification result. They performed this study on several state-of-the-art classification algorithms. (Dworak & Baranowski, 2022) attempted to adapt Grad-CAM Selvaraju et al. (2017) for object detection on point clouds by multiplying the bird's-eye view projection with an up-sampled CAM to produce high-resolution heatmaps. However, to the best of our knowledge, no work has been done to produce visual explanations for point cloud semantic segmentation architectures, making this our first attempt at doing so. We utilised RandLA-Net Hu et al. (2020), a popular and efficient encoder-decoder based architecture for semantic segmentation of large-scale point clouds, and the very popular and widely used SemanticKITTI Behley et al. (2019) dataset for experimentation. The output of each encoder and decoder is an activation layer that is used to produce heatmaps by per-point coloration. Relevant codes can be accessed at GeoAI and Abhishek.

## 2 Methodology & Experimentation

Consider a point cloud containing N number of points with the point space consisting of x-y-z coordinates and feature space consisting of attributes like Intensity, RGB values, etc., denoted as

$\mathcal{P} \in R^{N \times 3}$ and $\mathcal{F} \in R^{N \times d}$, respectively, where $d$ is the dimension of input feature space. Output of segmentation network are the logits for each point $i$, denoted by $l_i$. For any intermediate activation layer $A \in R^{M \times k}$ ($M$ downsampled points with $k$ feature dimension), gradient of logit $l_i$ for class $c$ w.r.t $A_k$ is given by:-

$$\frac{\partial l_i^c}{\partial A^k} \tag{1}$$

Eq. 1 is a measure of influence of the logit $l_i$ on the $k_{th}$ feature vector of $A$. Overall influence for N points can be obtained by summation of each gradient output:-

$$\frac{\partial l_1^c}{\partial A^k} + \frac{\partial l_2^c}{\partial A^k} + ... + \frac{\partial l_N^c}{\partial A^k} = \frac{\partial \sum_{i=1}^{N} l_i^c}{\partial A^k} \tag{2}$$

Above influence measure can be enhanced by performing pointwise addition with higher derivatives of Eq. 2. This leads to better localization and smoother heatmaps. Overall gradient influence $G^{k1}$ (using I+II+III order derivatives) aggregated over all $M$ points in the activation layer $A_k$ is given by:-

$$G^k = \sum_{j=1}^{M} \left[ \frac{\partial^3 \sum_{i=1}^{N} l_i^c}{\partial (A_j^k)^3} + \frac{\partial^2 \sum_{i=1}^{N} l_i^c}{\partial (A_j^k)^2} + \frac{\partial \sum_{i=1}^{N} l_i^c}{\partial A_j^k} \right] \tag{3}$$

Final heatmap (H) is obtained by matrix multiplication of $G^k$ with activation layer $A^k$ followed by ReLU operation (to highlight positive contributions only) and Min-Max normalization:

$$H = MinMax(ReLU(\sum_k G^k A^k)) \tag{4}$$

H acts as a scalar field with dimension $R^N$, which can be used for per point coloration. Figure 1 shows an example of heatmaps obtained for each activation layer of the RandLA-Net architecture.

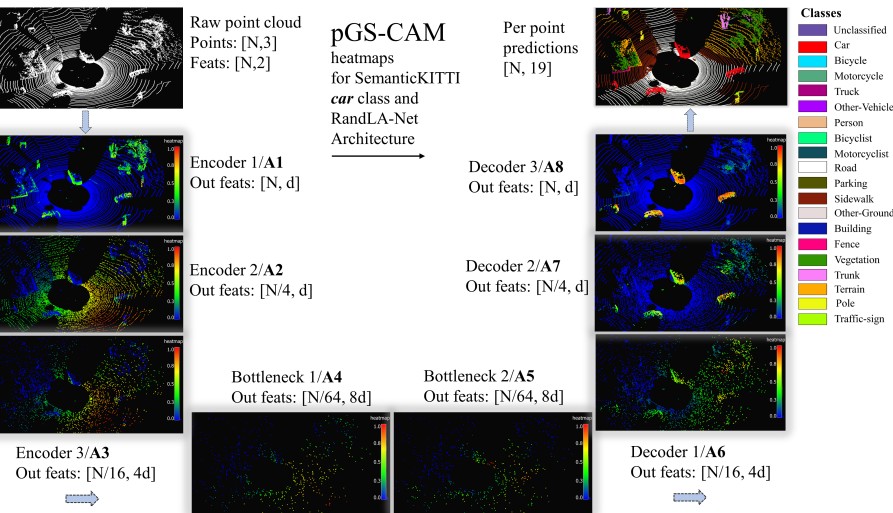

Figure 1: pGS-CAM heatmaps for the *car* class of SemanticKITTI. Heatmap at initial activation layer A1 exhibits edge-like structures with localization starting at A5. Heatmaps at A2-A4 highlights points (road, terrain) that might strongly impact prediction for *car* class. Heatmap at last activation layer A8 closely resembles the per point predictions for *car* class.

## 3  DISCUSSION AND FUTURE WORK

The proposed method produced promising outcomes, and a detailed analysis of pGS-CAM will be done in the future. Further investigation with metrics will be attempted for other popular semantic segmentation architectures to test the robustness of our method. This work has been implemented on MLS data; we also intend to test the method on airborne laser scanning (ALS) datasets to establish the proposed methodology as a benchmark.

---

[1]Higher order derivatives > 3 brought minimal changes hence we excluded them in the formulation.

URM STATEMENT

The authors acknowledge that at least one key author of this work meets the URM criteria of ICLR 2023 Tiny Papers Track.

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

## A  APPENDIX

### A.1  TESTING THE LOCALIZATION OF PGS-CAM HEATMAPS

To get a glimpse of pGS-CAM's faithfulness we can compute a localization metric for all classes of the dataset. One such metric is $IoU$ (Intersection over Union)/Jaccard Index commonly used in the point cloud segmentation tasks. For a class $c$ corresponding to an object, $IoU$ is calculated as:-

$$IoU = \frac{TP_c}{TP_c + FP_c + FNc} \tag{5}$$

We compute $IoU$ score between heatmap obtained at final decoder layer and per point predictions for class c. We do so by constructing a binary mask for heatmaps by only considering heatmap values greater than certain threshold. A higher $IoU$ score suggests that the model uses points corresponding to intended class c for its final predictions while lesser score suggest the involvement of points corresponding to other classes. We compare the $IoU$ score with slightly different formulations of our $G^k$ gradient influence 3 by varying number of higher order derivatives included in the summation operation. Table 1 demonstrate $IoU$ scores for 9 major classes of SemanticKITTI computed by taking mean $IoU$ of all point clouds scans from validation set ($8^{th}$ sequence of dataset). We consider

Table 1: Localization comparison using major SemanticKITTI classes for different $G_k$ modes along with segmentation *mIoU* performance for validation set. Higher $IoU$ represents better localization.

| $G^k$ modes | Car | Truck | Person | Road | Sidewalk | Building | Vegetation | Trunk | Terrain |
|---|---|---|---|---|---|---|---|---|---|
| I order | 51.4 | 10.2 | 20.4 | 78.4 | 36.1 | 50.4 | 58.8 | 35.6 | 56.6 |
| I+II order | 61.6 | 12.6 | 28.9 | 85.2 | 39.8 | 54.8 | 60.3 | 47.8 | 58.8 |
| **I+II+III order** | **66.4** | **13.9** | **34.4** | **87.9** | **41.5** | **57.5** | **64.3** | **50.6** | **65.1** |
| Segmentation *mIoU* | 93.0 | 67.7 | 48.8 | 91.4 | 76.5 | 85.5 | 84.2 | 58.5 | 73.6 |

3 modes of $G^k$ with I order, I+II order, I+II+III order derivatives included. Higher localization is observed for I+II+III mode when compared with I+II/I modes, however the impact mitigates as we keep adding higher order derivatives in chain. Note that localization value does not indicate effectiveness of the method. It is just that in case of low localization segmentation network use other (correlated) class points for its final prediction. However, a higher localization can be indicative of method's faithfulness in most situations.

## A.2 PGS-CAM INDICATES THE PREDICTION COMPLEXITY FOR DIFFERENT CLASSES

Usually, the classes having higher localization values ex- car, road, building, vegetation, etc., are relatively easier to predict while classes like truck, sidewalk, pole, etc., are harder and prone to errors. Table 1 provides insight for classes where the segmentation network struggles the most and uses nearby correlated points for final predictions. Classes with higher $IoU$ have higher number of instances in validation set, higher point density per scan and relatively simpler geometries making them easier for the network to learn.

## A.3 PGS-CAM OFFERS INSIGHTS INTO NETWORK MISCLASSIFICATIONS

Figure 3 showcase an example of misclassification where segmentation network assigns *person* semantic label to points surrounding the actual *person* object which can be seen as drawback of feature extraction. pGS-CAM heatmap (obtained at last decoder layer) assigns higher score to *person* class points than the neighboring points (red values for *person* object). We can conclude that network was able to identify the *person* object better than the nearby dissimilar points. Such information is useful for deriving valuable inferences during misclassifications and to further debug and modify the network.

## A.4 MORE QUALITATIVE RESULTS

Figure 2 demonstrates pGS-CAM explanations for KPConv Thomas et al. (2019) architecture and Paris-Lille3D Roynard et al. (2018) dataset *car* class. KPConv provides state of the art results for many datasets in point-based category. Paris-Lille3D is a dense dataset with 160 million points and 10 semantic classes. We observe similar edge-like structures as Figure 5-6 at initial activation layers A1-A2, however, unlike RandLA-Net architecture, the localization for KPConv architecture starts early at A3. Interestingly, KPConv encoders could identify car object pretty early within the network depth compared to RandLA-Net.

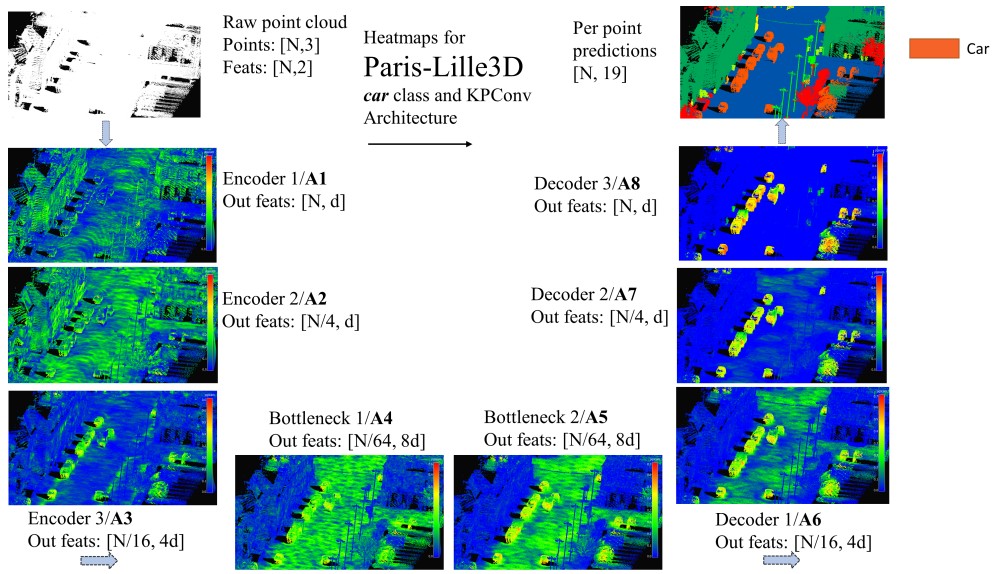

Figure 2: pGS-CAM heatmaps for the *car* class of Paris-Lille3D. Heatmap at initial activation layer A1-A2 exhibits edge-like structures with localization starting at A3. In KPConv, context points (road, terrain) are highlighted in the bottleneck layers (A4, A5). Heatmap at last activation layer A8 closely resembles the per point predictions for *car* class.

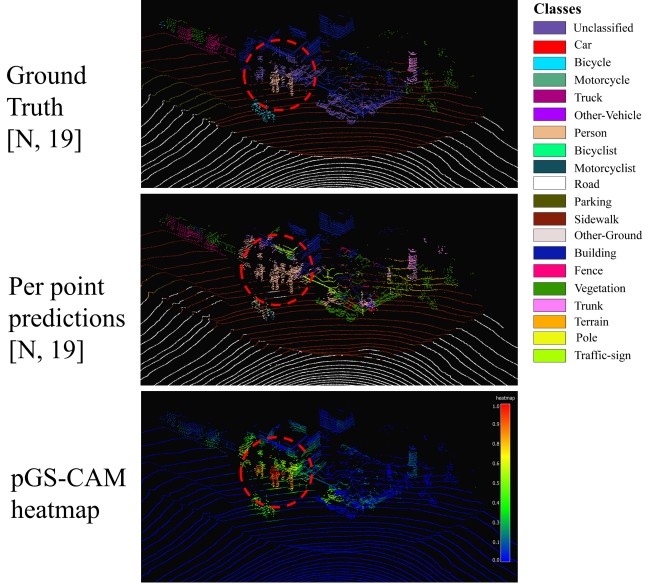

Figure 3: pGS-CAM heatmap for SemanticKITTI *person* class (enclosed within red dotted circle). Heatmap highlights points corresponding to *person* class more than surrounding points.

