# OpenReview forum: "pGS-CAM: Interpretable LiDAR Point Cloud Semantic Segmentation via Gradient Based Localization"
_ICLR.cc/2023/TinyPapers — Submitted to Tiny Papers @ ICLR 2023_

### Official Review · Reviewer_eMRP · 2023-03-30

**Confidence:** 3

**Summary Of Contributions:**

Reviews for Paper #86

**Rating:**

Clear, Correct, and Reproducible (CCR): a submission which meets the reviewing criteria

**Strengths And Weaknesses:**

This paper proposes a gradient analysis technique on models for point cloud semantic segmentation.

1. Clear formulations of the gradient importance and heatmaps are given and easy to follow;
2, FIgures provide intuitive demonstrations;
3, The technical novelty seems to be limited for mainstream venues. However, I think this paper meets the criteria in terms of clarify, correctness, and reproducibility. Thus, I vote for acceptance.



**Suggested Changes:**

1, Examination of the proposed methods on more segmentation models are encouraged.

---

> ### Author Response · Authors · 2023-05-27
> **Response to ICLR 2023 TinyPapers Reviewer eMRP**
>
> Thank you for your comments. We have added results for Paris-Lille3D and KPConv architecture to prove robustness for any dataset or segmentation architecture.

---

### Official Review · Reviewer_VNrU · 2023-04-01

**Confidence:** 4

**Summary Of Contributions:**

This tiny paper provides a method for utilizing GradCam for 3D Point Cloud Semantic Segmentation architecture

**Rating:**

Clear, Correct, and Reproducible (CCR): a submission which meets the reviewing criteria

**Strengths And Weaknesses:**

### Strengths:
- The paper is clear, well-motivated and proposes a solution to an important problem.

### Weaknesses:
1. The paper should mention the performance of the RandLA-Net semantic segmentation architecture. I want to see whether the GradCam performance in 3D point clouds affected by the underlying semantic segmentation module.



**Suggested Changes:**

Overall, I find this submission to be Clear, Correct and Reproducible. I also have some clarifications regarding the approach, stated below:
1. Why is Truck class harder and more prone to errors than any other class? Is it because the underlying segmentation module also had a low IoU segmentation performance for this class?
2. The authors mentioned "higher derivatives lead to better localization and smoother heatmaps at last decoder layers". Did the authors verify this using qualitative plots as well? Or was this claim made on just the quantitative performance reported in Table 1?

---

> ### Author Response · Authors · 2023-05-27
> **Response to ICLR 2023 TinyPapers Reviewer VNrU**
>
> Thank you for your comments and suggestions. We have highlighted the mIoU segmentation performance of RandLA-Net architecture on validation set in Table 1. Responses to suggested changes: -
> 1. mIoU performance for Truck class on validation set is 67.74 which is neither high nor low. Here, the reason for low localization could be due to the low instances of truck class in validation set. Model thus derives inferences from nearby context points. Poor localization does not necessarily mean low segmentation mIoU performance, but it is the general trend we observed.
> 2. Yes there are variations in qualitative plots for higher derivatives leading to low localization. The variations are generally subtle but noticeable in few cases.

---

### Author Response · Authors · 2023-05-27
**Response to the reviewers**

We acknowledge and thank the reviewers for their helpful comments and suggestions.
We have updated the submission and highlighted the following changes: -
1. Abbreviation mistakes has been corrected in abstract and main text.
2. Github link containing codes is provided in Introduction section.
3. Table 1 has been updated incorporating mIoU results for semantic segmentation using RandLA-Net architecture.
4. Qualitative plots are added in “More qualitative results” section highlighting results for Paris-Lille3D [A] and KPConv [B] architecture. This is to show that proposed methodology is robust for any dataset or segmentation architecture.
To maintain the brevity of this tiny paper we have included only necessary figures and plots.

[A] Xavier Roynard, Jean-Emmanuel Deschaud, and Franc ̧ois Goulette. Paris-lille-3d: A large and high-
quality ground-truth urban point cloud dataset for automatic segmentation and classification. The
International Journal of Robotics Research, 37(6):545–557, 2018.

[B] Hugues Thomas, Charles R Qi, Jean-Emmanuel Deschaud, Beatriz Marcotegui, Franc ̧ois Goulette,
and Leonidas J Guibas. Kpconv: Flexible and deformable convolution for point clouds. In
Proceedings of the IEEE/CVF international conference on computer vision, pp. 6411–6420, 2019.

---

### Comment · Area_Chair_DYMm · 2023-06-02
**ICLR Tiny Paper Archival (not in 2-pages limit)**

- This work " DISCUSSION AND FUTURE WORK" section is outside the 2-pages limit.
- The work contains the URM statement and is deanonymized.

---

### Comment · Area_Chair_DYMm · 2023-06-03
**ICLR Tiny Paper Archival**

This work meets the threshold for archival, contents the URM statement and is deanonymized.

---

### Meta-Review · Area_Chair_DYMm · 2023-04-09

**Recommendation:** Invite to present
**Confidence:** 4

**Metareview:**

This paper proposes a novel algorithm named pGS-CAM to explain and visualize point cloud semantic segmentation architectures via heatmaps. **The paper is clear, well-motivated, and proposes a solution to an important problem (interpretability)**. That is, improving the model's interpretability will further aid in neural network design by explaining the changes caused by the addition of each layer.

**Summary:**

This paper proposes a novel algorithm named pGS-CAM to explain and visualize point cloud semantic segmentation architectures via heatmaps. Reviewers suggest mentioning the performance of the RandLA-Net semantic segmentation architecture and to evaluate pGS-CAM on more segmentation models.

**Comments And Feedback To The Authors:**

I recommend to please not abbreviate "doesn’t" in scientific writing as in the abstract.


**Reason For Not Giving A Higher Recommendation:**

Although the authors mention open-sourcing the codes in the future, reproducibility was not possible to evaluate. For now, pGS-CAM has been evaluated in one dataset and the authors plan to analyze it deeply in future work as well.


**Reason For Not Giving A Lower Recommendation:**

The results presented for a single dataset are promising for improving the interpretability of deep learning models.

---

> ### Author Response · Authors · 2023-05-27
> **Response to ICLR 2023 TinyPapers Area Chair DYMm**
>
> Thank you for the encouraging review. Link for codes has been shared in the revised version.
> We have also corrected the abbreviation mistake in abstract.

---

### Decision · Program_Chairs · 2023-04-09

Invite to present